# Iatrogenic Esophageal Perforation in Premature Infants: A Multicenter Retrospective Study from Poland

**DOI:** 10.3390/children10081399

**Published:** 2023-08-16

**Authors:** Aleksandra Mikołajczak, Katarzyna Kufel, Joanna Żytyńska-Daniluk, Magdalena Rutkowska, Renata Bokiniec

**Affiliations:** 1Neonatal Department, Collegium Medicum of Cardinal Stefan Wyszynski University, 01-938 Warsaw, Poland; 2Department of Neonatology and Intensive Care, Medical University of Warsaw, Karowa 2, 00-315 Warsaw, Poland; kufel.k@gmail.com; 3Clinical Department of Neonatology, Central Clinic Hospital of Ministry of Interior and Administration, 02-591 Warsaw, Poland; joanna.daniluk@cskmswia.gov.pl; 4Neonatal and Intensive Care Department, Institute of Mother and Child, 01-211 Warsaw, Poland; magdalena.rutkowska@imid.med.pl

**Keywords:** esophageal perforation, preterm infants, feeding tube, chest X-ray, air leak

## Abstract

Greater awareness of possible iatrogenic esophageal perforation (EP) is needed. Though rare, EP is a legitimate health risk as it may lead to long-term morbidities. This study presents and discusses iatrogenic EP in a subset of preterm infants. Using radiographic images, we study and describe the consequences of the orogastric/nasogastric tube position (in radiographic images). We analyze the possible influence of histological chorioamnionitis on the development of esophageal perforation. This retrospective study examines the hospital records of 1149 preterm infants, 2009–2016, with very low birth weight (VLBW) and iatrogenic EP, comparing mortalities and morbidities between the two groups of preterm infants who had birth weights (BWs) of less than 750 g and were less than 27 weeks gestation age at birth: one group with iatrogenic esophageal perforation (EP group) and one group without perforation (non-EP group—the control group). Histopathological chorioamnionitis of the placenta showed no statistically significant differences between the groups. The only statistically significant difference was in the air leaks (*p* = 0.01). Three types of nasogastric tube (NGT) X-ray location were identified, depending on the place of the perforation: (1) high position below the carina mimicking esophageal atresia; (2) low, intra-abdominal; (3) NGT right pleura-directed. We also highlight the particular symptoms that may be indicative of EP due to a displacement of the nasogastric tube.

## 1. Introduction

Iatrogenic esophageal perforation (EP) is rare. It was historically prevalent, mostly in the Intensive Care Unit (ICU)/ Neonatal Intensive Care Unit (NICU) and continues to be so today. When it does occur, however, it is a life-threatening condition. Most at risk are premature infants of very low birth weight (VLBW). The mortality rate in neonates with EP varies from 20 to 30% [1].

Warden first reported iatrogenic esophageal perforation in a newborn in 1961, while Eklof did so in 1969 [2,3]. It is believed that almost all cases of esophageal perforation are iatrogenic. It occurs as a complication of necessary instrumentation. Causes of the injury might include the force exerted by repeated attempts at the time of og/ng tube placement, vigorous suctioning and repeated airway intubation [4,5]. According to Filippi’s study, the incidence of esophageal perforation is 1:25 in infants weighting less than 750 g and 1:124 of the preterm population weighing less than 1500 g [6]. 

The most common site of perforation is the cervical esophagus [6]. This is attributed to poor posterior esophageal musculature and delicate tissue in extremely-low-birth-weight (ELBW) neonates [7,8]. EP may be difficult to detect, but early recognition increases the chance of a successful outcome [9].

The aim of this study is to identify the risk factors, signs, symptoms and complications associated with iatrogenic EP in preterm infants admitted to the NICU and, therefore, to highlight the need for greater awareness of possible iatrogenic EP.

## 2. Materials and Methods

This study is a retrospective analysis of data collected for a multisite study to investigate iatrogenic perforation of the esophagus in preterm infants. All the hospital files of 1149 VLBW infants admitted to the two centers of the Neonatal Intensive Care Units in Warsaw between 2009 and 2016 were reviewed retrospectively. Preterm infants with EP were identified from the EHRs based on the diagnosis of esophageal perforation. Parents’ consent of infants with EP was obtained. The study was approved by the Bioethical Committee of Medical University of Warsaw, Poland (number AKBE/242/2019). 

Additionally, we compared mortalities and morbidities between two groups of preterm infants who had birth weights (BWs) of less than 750 g and were less than 27 weeks gestation age at birth: one group with iatrogenic esophageal perforation (EP group) and one group without perforation (non-EP group—the control group). 

For bronchopulmonary dysplasia (BPD), we used the new definition created by Jensen et al. in 2019, graded according to any respiratory support at 36 weeks post-menstrual age (PMA), regardless of oxygen concentration [10]. 

Sepsis is defined as a serious condition resulting from the presence of harmful microorganisms in the blood or other tissues. According to the guidelines of the Polish Society of Neonatology, sepsis is initially diagnosed based on the presence of at least two clinical symptoms (apnea, increased body temperature, symptoms of respiratory distress syndrome, hypoxia, reluctance to eat, somnolence or anxiety). The confirmation of the diagnosis is a positive culture [11].

The study focused on identifying the complications that could be attributed to esophageal perforation and their potential association with outcomes, i.e., length of stay, extended parenteral nutrition, delay in obtaining full enteral feedings, extended mechanical ventilation, sepsis and mortality.

Analyzed in the collected EP cases were the following factors: perinatal history, hospitalization course and patient’s outcome with regard to respiratory morbidities, such as air leak syndrome, pleural effusion, different radiological malposition of the og/ng tube, duration of oxygen therapy and the time to reach full enteral feeds. The frequency of early-onset sepsis and its association with perforation were also studied. A set of clinical symptoms typically accompanying EP was identified through detailed analyses of difficulties in the tube insertion, the character of stomach residues and the deterioration of respiratory function. 

The study also included a histopathological analysis of the placenta to find a possible causal relationship between esophageal perforation and intrauterine infection. 

### Statistical Analysis

In the first stage of the analysis, simple descriptive statistics were calculated for every variable. The relationship between qualitative variables was tested using the χ2 test and Fisher’s exact test, respectively, to the sample size. Quantitative variables were tested for probability distribution. Because the parameters did not follow a normal distribution, they were compared using the Wilcoxon rank-sum test. 

A *p*-value of <0.05 was considered statistically significant. The calculations were processed in SAS/STAT rel. 14.3.

## 3. Results

Thus, 10 of the 1149 preterm infants were diagnosed with iatrogenic esophageal perforation. Eight infants had birth weights (BWs) lower than 750 g and were less than 27 weeks gestation age at birth. Two infants had birth weights of 970 grams and 1100 grams and were born at 31 and 35 weeks, respectively. A group of 10 premature infants with EP (5 female/5 male) were studied. The Apgar score of the patients in the whole group with EP was low; median 3; range 1–7 points. The median birth weight and gestation age were 640 g (430–1100) and 24.45 (23.0–35.0) weeks. Four infants were small for their gestation age. The mode of delivery was as follows: six natural delivery, four caesarean section. Premature patients were managed by skilled medical staff.

Ten iatrogenic esophageal perforation cases were analyzed, giving an incidence of 1:115 (0.87%) in very-low-birth-weight infants, and the incidence reached 1:18 (5.3%) in neonates with BW under 750 g. Table 1 specifies the morbidities and mortalities of ten iatrogenic EP patients.

We compared mortalities and morbidities between two groups of preterm infants who had birth weights (BWs) of less than 750 g and were less than 27 weeks gestation age at birth: one group with iatrogenic esophageal perforation (EP group) and one group without perforation (non-EP group—the control group). The overall characteristics of the groups are shown in Table 2.

The diagnosis of esophageal perforation was confirmed via a radiographic image. All of the preterm neonates were intubated after birth and ventilated, and an orogastric (og)/ nasogastric (ng) tube was inserted as a regular procedure in the clinical course. A polyvinyl (PCV) feeding tube was used in all the patients in both groups. Typical placement of the ng/og tube involves measuring the distance between the sternal margin, the ear and the nose, followed by the insertion of the ng/og tube in the stomach, followed by checking the aspiration of the stomach juices’ digestive contents. The radiographic image is performed only in atypical cases, such as complications with the proper placement of the og/ng tube, blood aspiration instead of digestive contents, increased oral secretion, abdominal distension, poor feeding or sudden respiratory deterioration. 

The diagnosis of EP was confirmed via the malposition of the og/ng tube in plain chest X-rays radiographic images. In infants with EP, the chest radiographic image revealed three different malpositions of the og/ng tube. In five infants, the og/ng tube was located in the right pleural cavity (pleural position) (Figure 1), which is usually connected with the pneumothorax or the pneumothorax with pleural effusion, and a radiogram of another two patients indicated mimic esophageal atresia (high position) (Figure 2). In the last three infants, the og/ng tube was located in the abdomen (low position) (Figure 3). 

One patient had pneumothorax with pneumomediastinum, four infants had pneumothorax and another of them had milky pleural effusion. 

All infants with pneumothorax required urgent pleural decompression through chest drainage to relieve respiratory compromise. The aspiration of milky fluid during pleural decompression in one infant was analyzed and revealed a high triglyceride level. In all infants, the malpositioned polyvinyl tube was replaced by a properly inserted silastic og/ng tube and kept in place for another 7–10 days. 

Resolution of the perforation was confirmed only in two patients on a contrast study prior to restarting the feeds. Management included nil per os, lasting at least 7–10 days, total parenteral nutrition and intravenous broad-spectrum antibiotics to prevent mediastinitis or peritonitis. Patients required parenteral nutrition for a long time (median 32 days; range: 9–89) and, thus, the incidence of sepsis was 60% in the whole group with EP. Sepsis was determined by a positive blood culture. Six of eight infants < 750 grams had sepsis in the EP group. Among the microorganisms associated with sepsis were the following: Staphylococcus coagulase negative (5×); Candida albicans (1×). Respiratory failure was connected with prolonged mechanical ventilation (median 52 days; range: 19–116) and a high incidence of bronchopulmonary dysplasia (30%). 

Three infants died (30%) from prematurity-related sequelae (one infant following grade IV intraventricular hemorrhage and two due to sepsis with multi-organ failure). 

A comparative analysis of 8 neonates with esophageal perforation (EP group) and 142 neonates in the non-EP group born with a body mass (BM) of less than 750 g did not reveal differences in the gestational age and BW between the groups. The only statistically significant difference was in the air leaks (*p* = 0.01) between the EP group and the non-EP group. The morbidities and mortalities of the groups are shown in Table 3.

Overall, preterm infants with EP required a longer time to achieve full enteral feeding and to be discharged from hospital, but the differences were not statistically significant. Similarly, no statistically significant differences were observed in relation to the duration of oxygen therapy, mechanical ventilation and mortality rate.

Histological chorioamnionitis (HCA) was revealed in 48 cases, which accounted for 51.6% of the examined placentae. In the EP group, 4 (50%) placentae were examined, while in the non-EP group, it was 88 (61.9%). Histological chorioamnionitis was found in 2 patients in the EP group and 46 in the non-EP, which accounted for 50% and 52.2%, respectively. 

The results of the examined placentae indicated that HCA did not influence the frequency of esophageal perforation. In the whole group, congenital sepsis from a mother with histological chorioamnionitis was diagnosed in one patient only. The remaining cases of sepsis in both groups of patients were found to be secondary infections. BPD was diagnosed in 34 patients in the non-EP group. In 12 of them, histological chorioamnionitis-type changes were detected.

## 4. Discussion

This study highlights the necessity for greater awareness of possible iatrogenic esophageal perforation (IEP). The study shows that premature and small-for-gestational-age (SGA) infants are most at risk of iatrogenic esophageal perforation [12]. We analyzed IEP-associated morbidities and mortalities. According to the literature, the occurrence of IEP is rare, and it affects 2 to 4% of premature infants with BW < 750 g [13]. In our study, the rate was 5.3%.

Overall, we estimated the mortality index in premature infants with EP as being high. It was 37. 5% among premature infants with BWs of less than 750 g and less than 27 weeks gestation age while 30% in the whole group with EP. 

In our study, the incidence of SGA in the group with EP was 25% versus 11.27% in premature infants who had birth weights lower than 750 g, but against the whole sample of esophageal perforation, the incidence of SGA in this group reached 40%.The rarity of EP and differences in its clinical presentation heighten the need for increasing the awareness of IEP. Symptoms depend on the site of esophageal perforation and include acute respiratory deterioration, tachycardia and high fever but may sometimes manifest as subtle signs, such as blood aspiration, increased oral secretion, poor feeding and coughing [5]. Poor feeding is connected with symptoms, such as abdominal distention, visible bowels loops or with feeding intolerance defined as bilious residuals or pre-feed gastric residual volume of >50% in 2 ≥ consecutive feeds. High fever is a sign of possible sepsis or even mediastinitis connected with EP.

The pleural location of the og/ng tube is usually indicative of a sudden deterioration of the respiratory function, while the low and high position are largely asymptomatic (apart from absence of gastric contents and blood aspiration). Patients with abdominal esophageal perforations may present signs of peritonitis. Mediastinitis may develop as a result of leakage of the esophageal contents. 

Radiographs of the chest are useful in demonstrating the presence of the og/ng tube in situ and might reveal three different malpositions of the nasogastric tubes, typically in the right pleural cavity associated with right-sided pneumothorax or pneumomediastinum and sometimes pleural effusion. Common symptoms in this situation are connected with the deterioration of the respiratory status. A chest radiographic image may reveal excessive amounts of air in the mediastinum. The high position of the og/ng tube mimicking an esophageal atresia effect may be due to the presence of a mass created by a false passage of air or milk in the mediastinum [9,14]. The position of the top of the og/ng tube may be helpful in differential exclusion. In true esophageal atresia, it is usually above the bifurcation of the tracheae, while in the mimic esophageal atresia, it is located below the bifurcation. In doubtful cases, esophagoscopy can be conclusive [15]. The presence of gas along the og/ng tube may also be helpful in diagnosing EP. The third possible malposition of the og/ng tube involves the intra-abdominal location, which might be associated with the absence of gas in the intestines. It must be noted that a radiographic image cross-table lateral might, in this case, indicate where the tip of the og/ng tube would appear in the posterior/retroperitoneal space. This would be advisable for a more precise diagnosis. Another complication described by Sorens was mediastinal abscess requiring chest drain insertion [16]. Occasionally, patients with intra-abdominal EP might present symptoms of peritonitis or dysphagia and drooling [1]. In our study, like Elgendys’ study, pneumothorax and septicemia were frequent complications of esophageal perforation [17]. We diagnosed peritonitis in one patient and necrotizing enterocolitis in one patient. In one patient, cardiorespiratory decompression was connected to pleural milky effusion. Chylothorax should be considered and excluded upon differential diagnosis when milk is present in the pleural cavity and milky fluid can be aspirated from the right pleural cavity. Kairamkonda linked chylothorax with esophageal perforation, which may follow when the value of the laboratory fluid analyzed shows a triglyceride level of >1.1 mmol/l with absolute cell count > 1000 cells/mcl and lymphocyte fraction > 80% [18,19,20].

Due to the possibility of an easy misdiagnosis of esophageal perforation, there is a necessity to highlight the need for checking the position of the og/ng tube. Usually, a chest radiographic image is useful and allows the end of the og/ng tube to be located. But, in difficult situations, other methods of diagnosis should be applied to avoid misinterpretation. Maryuama described another possibility of detecting an og/ng tube malposition using sonography. The extra-esophageal location of the og/ng tube can be confirmed through an ultrasonography image with a water-soluble contrast injected into the tube showing gas in the pleural cavity [13]. Endoscopy in micro-preemies carries its own significant morbidity but may be a method for future review.

The principal rules to reduce potential morbidity should include prompt diagnosis and, thus, the possibility of early conservative management [1,6]. In particular, there seems to be a recent tendency to avoid surgery. The treatment of EP includes fasting, total parenteral nutrition and intravenous antibiotics [16]. Broad-spectrum antibiotics providing coverage for aerobes and anaerobes might be necessary to prevent mediastinal and peritoneal infections [14]. In contrast to Sorens’ study, we did not administer steroids or ranitidine [16]. Antifungal coverage is warranted in some cases, such as for patients who have received broad-spectrum antimicrobial drugs prior to perforation, patients who have received steroids and patients who have not improved after several days of appropriate antibacterial therapy.

The approach to enteral treatment feeding has changed over time. Over the past few years, a number of centres used enteral feeding of preterm infants with EP 4–5 days after injury, increasing it gradually. We determined that the perforation site is healed 7–10 days, after injury and maintenance of the gastric tube is required for 7–10 days. Due to this, the check-up radiographic image with esophageal contrast should be performed after 7–10 days. In the case of a contrast leakage beyond the lumen of the esophagus, the tube is inserted again for 7 days and again controlled. If healing is not observed, an operation is necessary. 

Our results seem to indicate that our patients who died due to sepsis associated with multiorgan dysfunction insufficiency should have been treated with broad-spectrum antibiotics for not only aerobes but anaerobes as well. Patients usually require urgent pleural decompression by means of chest drainage to relieve respiratory compromise.

In patients with BW > 750 g, iatrogenic esophageal perforation occurred in only two patients with SGA weighing 960 g (0th percentile) and 1100 g (9th percentile). This confirms that iatrogenic EP seems to be associated with ELBW and intrauterine growth restriction.

EP is an important iatrogenic complication that can occur in even the most experienced hands. Extra care is required during the instrumentation of the upper airway and the gastrointestinal tract.

This study has its limitations due to a number of factors: firstly, the relatively small number of observations in the EP group; secondly, the study was retrospective. It is possible that certain cases of esophageal perforation may have been missed in the absence of symptoms requiring chest radiograph. Thirdly, the histopathological examination of the placenta covered only 61.3% of the patients. 

The relatively small number of observations in the EP group as compared to the non-EP group might have contributed to a lower statistical validity of the results. Yet, given the rarity of the incidence of the complication, we chose to make an attempt at to statistically analyze them. In future, further assessing the complication in a larger group of patients may be beneficial. 

Esophageal perforation is rare. A thorough analysis of 10 cases is of great cognitive value for understanding its early and long-term consequences. The study’s aim was to provide a fundamental understanding of the symptoms, drivers and prevention methods for a rare issue. Further studies may enable more sophisticated treatment strategies as awareness increases.

## 5. Conclusions

Iatrogenic esophageal perforation can be potentially dangerous. Consequently, awareness of possible esophageal perforation and experience sharing among medical staff seem necessary. In infants with sudden respiratory deterioration and a feeding tube, it is crucial to very carefully assess the position of the tip of the og/ng tube via radiographic images. A chest radiographic image can prove very helpful in diagnostics. The inability to aspirate gastric contents or bloody secretion should alert the clinician to a potential esophageal perforation. Extremely preterm infants should, thus, be managed by experienced staff. In these premature neonates, we recommend using a polyurethane (PUR) or silastic orogastric tube with a soft end that can stay longer in the digestive tract. Polyvinyl feeding tubes are stiffer, less flexible and easier to insert but more traumatic. Silicone and PUR tubes are tissue-compatible, unlike PCV tubes [16].

Early and prompt diagnosis gives a chance for non-operative treatment of EP. Prompt diagnosis and the option of conservative management could lead to a reduction in potentially significant morbidity. 

Radiographic image examination revealed symptoms potentially indicative of EP following a nasogastric tube displacement, correlating with a specific og/ng tube misplacement, which is worth highlighting as important for early EP detection.

## Figures and Tables

**Figure 1 children-10-01399-f001:**
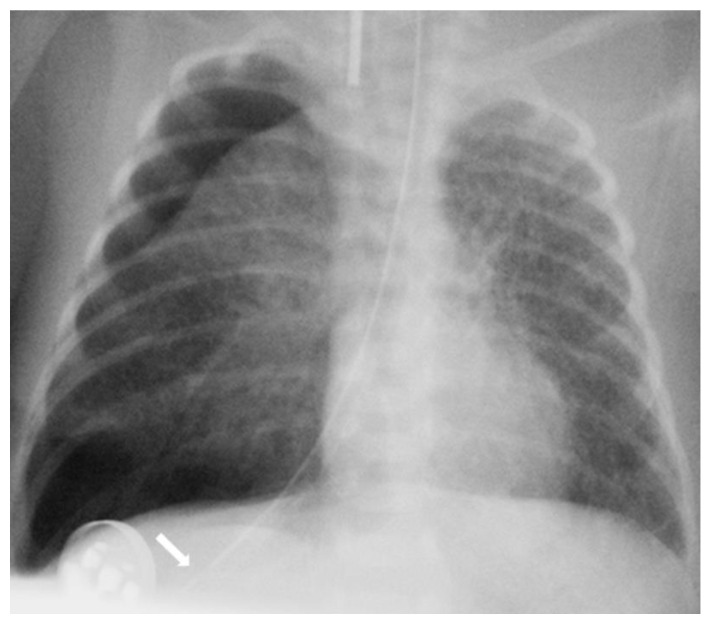
Large right tension pneumothorax with a mediastinal shift. A feeding tube is directed into the right pleura (arrow).

**Figure 2 children-10-01399-f002:**
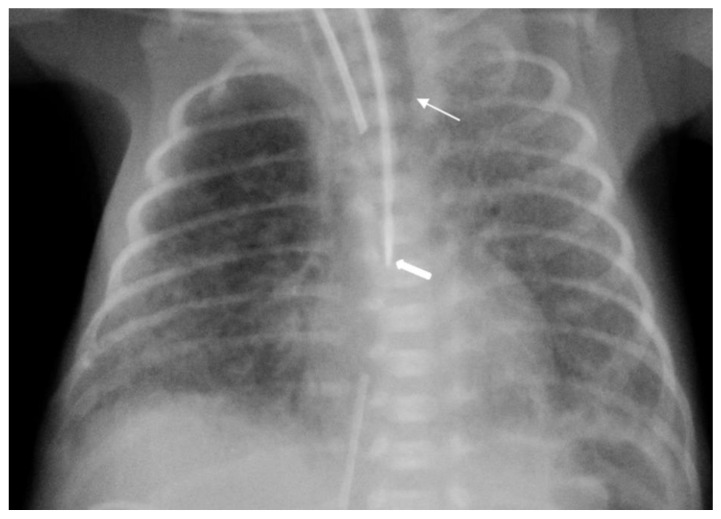
Supine AP view of the chest and upper abdomen. The nasogastric tube tip is in a high position below the carina (arrowhead) and can mimic esophageal atresia. Gas outside the esophagus resulted from its rupture (arrow). The umbilical line is present.

**Figure 3 children-10-01399-f003:**
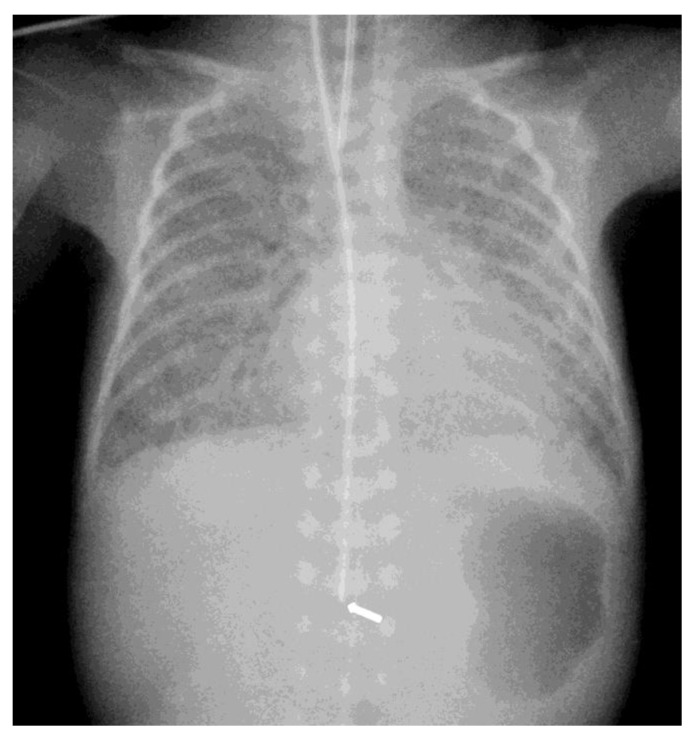
Intra-abdominal location of the NGT. The arrow indicates the position of the nasogastric tube tip in the abdominal cavity.

**Table 1 children-10-01399-t001:** Clinical data of ten patients with EP.

	Sex	GA (w^d^)	BW (g)	Age (day)	HCA	X-ray	Air Leaks	PN	Sepsis	ABX	MV	O2 Thx	BPD	Comorbi- Dities	Dis- Charge	Out Come
1	M	26^0^	430	1	(-)	PTX	Y	NA	Y	Vanc Gent Merop	NA	NA	NA	IUGR RDS, PDA Multi-organ Failure	NA	Death on D16
2	W	23^0^	500	3	NA	High *	N	49	Y	Vanc Clind Gent	71	116	Y	PDA IVH III	152	Surv
3	M	23^0^	630	1	(+)	PTX	Y	NA	Y	Vanc Gent	NA	NA	NA	PDA NEC Multi-organ Failure	NA	Death on D17
4	W	24^0^	650	1	(-)	High *	N	850	Y	Vanc Netil Merop Amfot B	89	108	Y	NEC fungemia renal- failure	180	Surv
5	M	23^1^	660	2	(-)	Low **	N	NA	N	Amp Gent	NA	NA	NA	IVH RDS NEC III grade	NA	Death on D11
6	W	26^0^	535	1	NA	Low **	N	23	N	Tazob Gent	9	52	N	IUGR RDS Pneum	99	Surv
7	W	24^3^	590	3	NA	Low **	N	333	Y	Tazob Gent	37	NA	Y	RDS PDA Ileum Perfor	141	Surv
8	W	24^3^	650	7	NA	PTX	Y	32	Y	Merop	43	52	N	RDS NEC	125	Surv
9	M	31^0^	1100	7	NA	PTX	Y	25	N	Vanc Amik	16	19	N	IUGR NEC IVH III	79	Surv
10	M	35^0^	970	3	(+)	PMD	Y	20	N	Vanc Merop Gent	15	33	N	IUGR	87	Surv

BPD = bronchopulmonary dysplasia; BW = birth weight; HCA = Chorioamnionitis; PN = Parental Nutrition DOL = day of life; GA W^d^ = gestational age; weeks^days^; ABX = Antibiotics; O2 THx = Oxy Therapy; IUGR= intrauterine growth restriction; IVH = intraventricular hemorrhage; MV = mechanical ventilation; NA = Not applicable; NEC = necrotizing enterocolitis; PDA = patent ductus arteriosus; ND = Non Aplicable; PMD = pneumomediastinum; PTX = pneumothorax; RDS = respiratory distress syndrome; Surv = Survival; High * =location high in mediastinum of feeding tube—mimic an esophageal atresia; Low ** =intraabdominal location of feeding tube; Vanc = Vancomycin; Gent = Gentamicin; Amik = Amikacin; Merop = Meropenem; Netyl = Netilmicin; Taz = Tazobactam; Clind = Clindamicin.

**Table 2 children-10-01399-t002:** Demographic characteristics of the study cohort.

Characteristics	EP Group *n* = 8	Non-EP Group *n* = 142	*p*
Gestational age, weeks, median (range)	24.21 (23–26)	24.0(23.0–26.9)	0.69 ^a^
Birth weight, g, median (range)	610 (430–660)	640 (410–748)	0.12 ^a^
Sex, male, *n* (%)	3 (37.5)	59 (41.6)	1.00 ^b^
Cesarean section, *n* (%)	2 (25.0)	80 (56.3)	0.14 ^b^
Small for gestation age, *n* (%)	2 (25)	16 (11.3)	0.25 ^b^

Data are expressed as median and ranges for nonparametric outcomes or *n* (%); ^a^ Wilcoxon test; ^b^ Fisher test.

**Table 3 children-10-01399-t003:** Comparison of morbidity and mortality between EP group and non-EP group.

Morbidities & Mortalities	EP Group *n* = 8	Non-EP Group *n* = 142	*p*
Achievement of enteral feeding (days), median (IQR)	49.0 (32.0–333.0)	29.0 (21.5–40.5)	0.06 ^a^
Duration of oxygen therapy (days), median (IQR)	90.0 (65.0–108.0)	72.0 (60.5–89.0)	0.42 ^a^
Mechanical ventilation (days), median (IQR)	43.0 (37.0–71.0)	35.0 (23.0–46.0)	0.23 ^a^
Discharge (days), median (IQR)	141.0 (125.0–152.0)	116.0 (102.0–136.0)	0.17 ^a^
Air leaks, *n* (%)	3.0 (37.5)	7.0 (4.9)	0.01 ^b^
Sepsis, *n* (%)	6.0 (75.0)	55.0 (38.7)	0.06 ^b^
Brochopulmonary dysplasia, *n* (%)	3.0 (60.0)	34.0 (38.6)	0.38 ^b^
Death, *n* (%)	3.0 (37.5)	54.0 (38.0)	1.00 ^b^
Chorioamnionitis *n* (%)	2.0 (50.0)	46.0 (52.3)	0.61 ^b^

Data are expressed as median and interquartile ranges (IQR) for nonparametric outcomes and % for categorical outcomes. For EP group: BPD should stay 3 (60%) due to the fact that at the time of assessment at 36 weeks GA only five patients of EP group at less than 27 weeks GA and less than 750 g were alive. ^a^ Wilcoxon test; ^b^ Fisher test.

## Data Availability

The data presented in this study are available on request from the corresponding author.

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
