# Peer review of "Iatrogenic Esophageal Perforation in Premature Infants: A Multicenter Retrospective Study from Poland"

_children, 2023, doi:10.3390/children10081399_

Round 1
Reviewer 1 Report
This is an important study contributing to the literature on the clinical and radiological features of iatrogenic esophageal perforation, which is rare in premature infants. One of the most important problems of the study is the small number of case groups. This may have caused the statistical results to be interpreted differently. It is important that this issue and its implications have been discussed by the authors in the discussion section. In addition, the article needs to be revised in terms of English.
The article needs to be revised in terms of English.
Author Response
The authors would like to express their gratitude to all three reviewers for their comments regarding the paper entitled “Complications due to iatrogenic esophageal perforation in premature infants”. Thank you for the time devoted to this endeavour and for your valuable observations and remarks.
I updated the parts of the study to reflect the reviewers' remarks
Response for Reviewer 1
Following the Reviewer’s suggestions:
The article was revised in terms of English.
I reviewed the entire manuscript for grammatical errors.
The quality of the manuscript was improved by native English speaker.
Reviewer 2 Report
In this paper, the authors have shared the results of a retrospective study, describing the risk factors and complications of iatrogenic esophageal perforation in 2 NICUs in Poland. Esophageal perforation is a rare but life-threatening condition in premature infants, however, very few papers have described the epidemiology and complications associated with it, and more so, there have been no such studies from Poland. This study, however, requires significant revision before it can be accepted for publication.
Abstract:
Line 18: ... may generate developmental problems- please delete and replace by: may lead to long term morbidities.
Line 20: Delete: 'studies or'
1. Introduction:
Line 34: Please state the full form of ICU and NICU before using abbreviations in the subsequent text
Lines 38-39: Eklof and Girdany..... hypopharynx: Warden first reported iatrogenic esophageal perforation in the newborn in 1961 compared to Eklof in 1969. Please correct and add the citation below
WARDEN HD, MUCHA SJ. Esophageal perforation due to trauma in the newborn. A case report. Arch Surg. 1961;83:813-815. doi:10.1001/archsurg.1961.01300180013003
Line 39: It is believed that all cases of esophageal perforation are iatrogenic: Please reframe as: "It is believed that almost all cases of esophageal perforation are iatrogenic.
Lines 49-55: The authors state the aim of the study to identify the signs and symptoms of EP but the manuscript is titled "complications of iatrogenic esophageal perforation". Please appropriately adjust Aim or title to descibe study objectives.
I suggest the following wording:
The aim of this study is to identify the risk factors, signs and symptoms and the complications associated with iatrogenic EP in preterm infants admitted to the NICU, and therefore to highlight the need for greater awareness of possible iatrogenic EP.
We estimated the frequency of EP-51 associated morbidities and mortality. We compared the outcome of preterm infants with 52 and without esophageal perforation in the <27 gestation week and BM <750g group. 53
The study analyzed the influence of histological chorioamnionitis on the incidence of 54 esophageal perforation.
I also suggest that the title be changed to: Iatrogenic esophageal peforation in premature infants: A multicenter retrospective study from Poland, however, I leave this upto the authors' discretion.
2. Materials and Methods:
- Line 62: How was parental consent obtained in this retrospective study? Was Informed consent waived by the bioethical committee or were the parents of infants with EP contacted before data collection? Additionally, were the parents of the patients with BW <750 grams used as controls also contacted?
-Lines 67-68:
The characteristics of the EP and Non-EP group in table 1 should be included in the results section and not in methods.
-- the definitions of various variables/outcomes studied must be menitioned in the methods section
3. Results: Authors should make edits to the results section to avoid repetition of data already in the table.
-Lines 111-113: The text is repetitive. Can be mentioned as median with range if necessary.
-Line 113: Please explain by what it meant by diagnosis and radiographic image coincided in time?
Lines 118-121: I understand that these were the clinical symptoms associated with esophageal perforation. Please mention the frequency of all the symptoms associated with EP.
Line 118: Would suggest replacing 'juices' With 'contents'.
Line 114: Please use full form of og and ng 'orogastric (og)/nasogastric (ng)' as it is the first time these abbreviations are being used.
Lines 142-144: Please explain "properly" inserted tube. Was the tube inserted under fluoro guidance?
Table 3: The numbers and percentage of some variables do not match: For example: For EP group: BPD should be 3 (60 37.5), Sepsis: 7 (75 87.5), etc.
For chorioamnionitis, please mention the number of placentae in both groups that were examined, otherwise the percentages would not matched.
For Air leaks, the p value while appropriately mentioned in the text as significant, is incorrect in the table and shown as non-significant. Please correct.
Lines 162-164: The calculations.... not statistically significant: please delete.
Also, how was sepsis determined. The authors mentioned 7 of 8 infants <750 grams having sepsis in the EP group. Was there a positive blood culture? If yes, what was the microorganism associated with sepsis.
4. Discussion:
Lines 183-184: The current study has shown no difference in babies with and without EP in terms of SGA status. Please correct it. None of the papers referenced (3, 9) have stated that EP is higher in SGA babies.
Overall the discussion appears a bit repetitive and slightly unorganized. Please correct so as to have a clear flow of thoughts.
- Please provide references for lines 251-256.
- Lines 257-259: While it is reasonable to cover anaerobes in infants with signs of sepsis following EP, were the blood cultures for EP patients positive for anaerobes?
- 261-263. Occurrence of EP in 2 SGA infants does not 'CONFIRM' anything. While authors may mention that SGA infants 'MAY' be at higher risk, please refrain from making definitive statements of association without statistical proof.
- Lines 264-265: Tiny premature.... increased risk as well: Please delete
-Line 265: Delete unpublished
Thank you for including the limitations of the study. Another limitation of this study is that this was a retrospective study. It is possible that certain cases of esophageal perforation may have been missed in the absence of symptoms requiring chest radiograph.
-Lines 280-285: Repetitive, please condense and delete the last two lines.
5. Conclusion:
- Lines 294-295: Is there any evidence which suggests that silicone tubes decrease the risk of EP compared to polyurethane? If yes, please provide reference. Was there a difference in the type of tubes used in the patients with EP vs no EP in this study? If yes, please include in results and methodology section.
Additionally, the references cited are not uptodate and many recent studies are not mentioned. Please add in discussion and cite results from the following recently published studies on neonatal esophageal perforation, while comparing the results of the present study
1. Sorensen E, Yu C, Chuang SL, et al. Iatrogenic Neonatal Esophageal Perforation: A European Multicentre Review on Management and Outcomes. Children (Basel). 2023;10(2):217. Published 2023 Jan 26. doi:10.3390/children10020217
2. Adel MG, Sabagh VG, Sadeghimoghadam P, Albazal M. The outcome of esophageal perforation in neonates and its risk factors: a 10-year study. Pediatr Surg Int. 2023;39(1):127. Published 2023 Feb 15. doi:10.1007/s00383-023-05417-x
3. Elgendy MM, Othman H, Aly H. Esophageal perforation in very low birth weight infants. Eur J Pediatr. 2021;180(2):513-518. doi:10.1007/s00431-020-03894-z
4. Hesketh AJ, Behr CA, Soffer SZ, Hong AR, Glick RD. Neonatal esophageal perforation: nonoperative management. J Surg Res. 2015;198(1):1-6. doi:10.1016/j.jss.2015.05.018
Please add additional references as appropriate and include in discussion.
There are several grammatical errors in the manuscript. Please review the entire manuscript for grammatical errors.
Would recommend enlisting help of personnel proficient in English language to improve the quality of the manuscript.
Author Response
The authors would like to express their gratitude to all three reviewers for their comments regarding the paper entitled “Complications due to iatrogenic esophageal perforation in premature infants”. Thank you for the time devoted to this endeavour and for your valuable observations and remarks.
I updated the parts of the study to reflect the reviewers' remarks
Response for reviewer 2
Following the Reviewer’s suggestions the authors:
1.Abstract:
Line 18: I replaced: “may generate developmental problems” to may lead to long term morbidities.
Line 20: I deleted “studies or”.
- Introduction:
Line 34: I used the full term of ICU and NICU before using the abbreviations in the subsequent text.
Lines 38-39: I changed the sentence and added information and the citation: “Warden first reported iatrogenic esophageal perforation in the newborn in 1961 compared to Eklof in 1969”.
Line 39: The sentence “It is believed that all cases of esophageal perforation are iatrogenic” I reframed as “It is believed that almost all cases of esophageal perforation are iatrogenic”.
Line 49-55: Following the reviewer suggestions I adjusted the Aim to describe study objectives:
“The aim of this study is to identify the risk factors, signs and symptoms and complications associated with iatrogenic EP in preterm infants admitted to the NICU, and therefore to highlight the need for greater awareness of possible iatrogenic EP”.
I deleted the sentences in lines 49-55.
I changed the title to: Iatrogenic esophageal perforation in premature infants: A multicenter retrospective study from Poland.
3.Materials and Methods:
Line 62: The parents of infants with EP were contacted before data collection, and the parental consent was obtained. The parents of the patients with BW <750g of non-EP group used as controls were not contacted. The Bioethical Committee stated that the study is not a medical experiment.
Lines 67-68:
I included the characteristics of the EP and Non-EP group in the results section of table 1,and I deleted from the methods section. I changed the number of the table: 1 to 2; and 2 to 1.
I added the definitions of various variables/outcomes studied in the methods section
Sepsis is defined as a serious condition resulting from the presence of harmful microorganisms in the blood or other tissues. According to the guidelines of the Polish Society of Neonatology, sepsis is initially diagnosed based on the presence of at least two clinical symptoms (apnea, increased body temperature, symptoms of respiratory distress syndrome, hypoxia, reluctance to eat, somnolence or anxiety). The confirmation of the diagnosis is a positive culture.
The new definition for bronchopulmonary dysplasia (BPD) created by Jensen et al. in 2019 is graded according to any respiratory support assessed at 36 weeks PMA regardless of oxygen concentration.
We assessed histological chorioamnionitis.
Jensen, E.A.; Dysart, K.; Gantz, M.G.; McDonald, S.; Bamat, N.A.; Keszler, M.; Kirpalani, H.; Laughon, M.M.; Poindexter, B.B.; Duncan, A.F.; Yoder, B.A. The diagnosis of Bronchopulmonary Dysplasia in Very Preterm Infants. An Evidence-based Approach. Am. J. Respirat. Crit. Care Med. 2019, 751-759. http://doi.org/10.1164/rccm.201812-2348OC
Borszewska-Kornacka, M.K.; GulczyÅ„ska , E.; Helwich, E.; Królak-Olejnik, B.; Lauterbach, R.; Maruniak-Chudek, I.; Szczapa, T. Standardy Opieki Medycznej Nad Noworodkiem w Polsce. Zalecenia Polskiego Towarzystwa Neonatologicznego. 4thed.; Medi Press: Warsaw. 2021, 318-320.
- Results: Authors should make edits to the results section to avoid repetition of data already in the table.
Lines 111-113: I deleted the repetitive text.
Line 113: I corrected the sentence to “The radiographic image confirmed the diagnosis of esophageal perforation”.
Lines 118-121: Among the symptoms associated with EP acute respiratory deterioration occurred more often (five/ten); the incidence of respiratory deterioration and poor feeding like abdominal distension, residual digestive contents was three/ten; only poor feeding with two/ten; subtle signs like bloody aspiration instead of digestive contents occurred with frequency of two/ten.
Line 118: I replaced “stomach juices” with “digestive contents”.
Line 114: I corrected abbreviations and used the full form of og and ng 'orogastric (og)/nasogastric (ng)' as it is the first time these abbreviations are being used.
Lines 142-144: Fluoro guidance wasn’t used to insert the tube "Properly". A properly inserted tube should be located in the lumen of the stomach. The radiographic image confirmed the tube was positioned correctly.
Table 3: I corrected the numbers and percentage of sepsis on Table 3: 6 (75%). For EP group: BPD should stay 3 (60%) due to the fact that at the time of assessment at 36 weeks GA only five patients of EP group at less than 27 weeks GA and less than 750g were alive.
For the whole group with esophageal perforation the indication for sepsis is six/ten (60%).
For chorioamnionitis, I mentioned the number of placentae in both groups that were examined, the percentages are accounted as a rate of positive results of HCA of placentas to all examined placentas.
Histological chorioamnionitis was revealed in 48 cases which accounted for 51.6% of the examined placentae. In the EP group, two/four (50%) placentae were positive while in the non-EP group 46/88 (52,3%).
Resullts of examined placentas indicated that HCA didn’t infuence the frequency of esophageal perforation).
For Air leaks, I corrected the p value in the table for significant p=0.01.
Lines 162-164: I deleted the sentence “The calculations.... not statistically significant”.
Sepsis was determined by a positive blood culture. There were 6 of 8 infants <750 grams having sepsis in the EP group. Among the microorganisms associated with sepsis were: Staphylococcus coagulase negative (5x); Candida albicans (1x).
5.Discussion:
Lines 183-184 I corrected the references for: Al-Khawahur, H.A.; Al-Salem, A.H. Iatrogenic perforation of the esophaus. Saudi Med J. 2002, 23(6):732-4.
I rearranged the discussion slighty.
I added references:
- Sorensen, E.; Yu, C.; Chuang, S.L. et al. Iatrogenic Neonatal Esophageal Perforation: A European Multicentre Review on Management and Outcomes. Children (Basel). 2023;10(2):217. Published 2023 Jan 26. doi:10.3390/children10020217
- Elgendy, M.M.; Othman, H.; Aly, H. Esophageal perforation in very low birth weight infants. Eur J Pediatr. 2021;180(2):513-518. doi:10.1007/s00431-020-03894-z
- Hesketh, A.J.; Behr, C.A.; Soffer, S.Z.; Hong, A.R.; Glick, R.D. Neonatal esophageal perforation: nonoperative management. J Surg Res. 2015;198(1):1-6. doi:10.1016/j.jss.2015.05.018
Lines 257-259 It is reasonable to cover anaerobes with antibiotics due to their presence in the gastrointestinal tract and the possibility of migration to other places such as retroperitoneal or mediastinum and easy multiplication.
Lines 264-265 I deleted
Line 265 I deleted “unpublished”
Regarding limitations I added “The study has its limitations due, firstly, to the relatively small number of observations in the EP group; Secondly, the study was retrospective; Thirdly, fact that the histopathological examination of the placenta covered only 61.3% of the patients.
Lines 280-285 I corrected and deleted the last two lines.
I changed to:
Esophageal perforation is rare. A thorough analysis of 10 cases is of great cognitive value for understanding its early and long term consequences. The study’s aim was to provide a fundamental understanding of the symptoms, drivers and prevention methods for a rare issue. Further studies may enable more sophisticated treatment strategies as awareness
6.Conclusion:
Lines 294-295 I added information regarding the PUR tube: “Silicone as well as polyurethane (PUR) tube are tissue compatible against polyvinyl chlorid (PVC) tube[Sorensen]. The material is soft and is easier for neonatal application.”
I reviewed the entire manuscript for grammatical errors.
The quality of the manuscript was improved by native English speaker.
Round 2
Reviewer 2 Report
Thank you for incorporating my comments and revising the manuscript accordingly. There are minor revisions that still need to be made before the manuscript is accepted. 1. Author's comments for Table 3: I corrected the numbers and percentage of sepsis on Table 3: 6 (75%). For EP group: BPD should stay 3 (60%) due to the fact that at the time of assessment at 36 weeks GA only five patients of EP group at less than 27 weeks GA and less than 750g were alive. -- Please mention this in the captions below the table 2. Line 203: Please add '(HCA)' after histological chorioamnionitis as the term is being used the first time here. 3. Please insert in the results section: Sepsis was determined by a positive blood culture. There were 6 of 8 infants <750 grams having sepsis in the EP group. Among the microorganisms associated with sepsis were: Staphylococcus coagulase negative (5x); Candida albicans (1x). (This is important information) 4. Line 295: Replace 'didn't' with 'did not'. 5. Line 325: Include after retrospective: It is possible that certain cases of esophageal perforation may have been missed in the absence of symptoms requiring chest radiograph. 6. Line 326: Delete: fact that
no
Author Response
Dear Reviewer 2!
Thank you for the comments and remarks. I revised the manuscript following the Reviewers' remarks.
Response for reviewer 2
- I added the explanation about assessment of BPD in the captions below the table 2: “BPD should stay 3 (60%) due to the fact that at the time of assessment at 36 weeks GA only five patients of EP group at less than 27 weeks GA and less than 750g were alive”.
2.In line 208 I added “(HCA) after histological chorioamnionitis as the term is being used the first time here.
3.Line 183-186 - I inserted in the results section: Sepsis was determined by a positive blood culture. There were 6 of 8 infants <750 grams having sepsis in the EP group. Among the microorganisms associated with sepsis were: Staphylococcus coagulase negative (5x); Candida albicans (1x).
4.Line 301: I replaced 'didn't' with 'did not'.
- Line 331-333: I included after retrospective: It is possible that certain cases of esophageal perforation may have been missed in the absence of symptoms requiring chest radiograph.
- Line 333: I deleted: fact that